# Unsupervised Speech Recognition via Segmental Empirical Output Distribution Matching

**Chih-Kuan Yeh**[*]
Machine Learning Department
Carnegie Mellon University
Pittsburgh, PA 15213, USA
cjyeh@cs.cmu.edu

**Jianshu Chen, Chengzhu Yu & Dong Yu**
Tencent AI Lab
Bellevue, WA 98004, USA
{jianshuchen,czyu,dyu}@tencent.com

## Abstract

We consider the problem of training speech recognition systems without using any labeled data, under the assumption that the learner can only access to the input utterances and a phoneme language model estimated from a non-overlapping corpus. We propose a fully unsupervised learning algorithm that alternates between solving two sub-problems: (i) learn a phoneme classifier for a given set of phoneme segmentation boundaries, and (ii) refining the phoneme boundaries based on a given classifier. To solve the first sub-problem, we introduce a novel unsupervised cost function named Segmental Empirical Output Distribution Matching, which generalizes the work in (Liu et al., 2017) to segmental structures. For the second sub-problem, we develop an approximate MAP approach to refining the boundaries obtained from Wang et al. (2017). Experimental results on TIMIT dataset demonstrate the success of this fully unsupervised phoneme recognition system, which achieves a phone error rate (PER) of 41.6%. Although it is still far away from the state-of-the-art supervised systems, we show that with oracle boundaries and matching language model, the PER could be improved to 32.5%. This performance approaches the supervised system of the same model architecture, demonstrating the great potential of the proposed method.

## 1 Introduction

Over the past years, the performance of automatic speech recognition (ASR) has been improved greatly and the recognition accuracy in certain scenarios could be on par with human performance (Xiong et al., 2016). Most of the state-of-the-art ASR systems are constructed by training deep neural networks on large-scale labeled data using *supervised* learning (Hinton et al., 2012; Dahl et al., 2012; Xiong et al., 2016; Graves et al., 2013; 2006; Graves, 2012); they rely on a large number of human labeled data to train the recognition model. In this paper, we are working towards the grand mission of training speech recognition models without any human annotated data. Such an approach could potentially save a huge amount of human labeling costs for developing ASR systems by leveraging massive unlabeled speech data. It is especially valuable when developing ASR systems for low-resource languages, where labeled data are more expensive to obtain.

Specifically, we consider the phoneme recognition problem, for which we learn a sequential classifier that maps speech waveform into a sequence of phonemes. In our unsupervised learning setting, the learning algorithm can only access (i) the input speech acoustic features, and (ii) a pretrained phoneme language model (LM). There is no human supervision presented to the algorithm at any level; that is, we do not provide any (frame-level) label for input samples, nor do we provide any (sentence-level) transcription for input utterances. The language model could be trained from a separate (text) corpus in an unsupervised manner with the help of a pre-defined lexicon[1].

There have been some recent successes in developing fully unsupervised method for neural machine translation (Artetxe et al., 2018; Lample et al., 2018) and sequence classifications (Liu et al.,

---

[*]The work was done during an internship at Tencent AI Lab, Bellevue, WA.

[1]A lexicon in ASR is a pre-defined dictionary that maps word sequences into phoneme sequences.

2017). However, different from these problems, speech recognition problem has segmental structures that impose unique challenges for developing unsupervised learning algorithms. First, each phoneme generally consists of a segment of consecutive input samples (frames) that are associated to the same phoneme label. Second, the lengths and the boundaries of these segments are usually unknown a priori. For this reason, we could not directly apply the previous techniques to develop unsupervised ASR algorithms. To address the first challenge, we develop a novel unsupervised learning cost function for ASR systems by extending the Empirical Output Distribution Matching (Empirical-ODM) cost in (Liu et al., 2017) to segmental structures. The key ideas of our Segmental Empirical-ODM are: (i) the distribution of the predicted outputs across consecutive segments shall match the phoneme language model and (ii) the predicted outputs within each segment should be equal to each other as they belong to the same phoneme. This cost function allows us to learn the classifier without labeled data for a given set of phoneme segmentation boundaries. To address the second challenge, we develop a novel unsupervised approach to estimate (and refine) the segmentation boundaries using the current classification model. Our algorithm alternates between these two steps of learning classifier and estimating the boundaries to successively improve the performance of each other. Therefore, unlike previous works in (Liu et al., 2018), which relies on an oracle or forced alignment methods to obtain the phoneme segmentation boundaries, our method is fully unsupervised in both segmentation and classification. Furthermore, we also adapt the semi-supervised HMM learning technique (Zavaliagkos et al., 1998; Kemp and Waibel, 1999; Nallasamy et al., 2012) to our unsupervised setting to further improve the performance. In our experiments on TIMIT phoneme recognition task, our unsupervised learning method achieves a promising phone error rate (PER) of $41.6\%$. To our best knowledge, this is the first empirical success of a fully unsupervised speech recognition that does not use any oracle segmentation or labels. Furthermore, when the oracle phoneme segmentation boundaries are given (similar to the setting in Liu et al. (2018)), our method achieves a PER of $32.5\%$ with matching language model, which approaches supervised learning with the same model architecture, demonstrating a great potential of our method.

## 2 FULLY UNSUPERVISED SPEECH RECOGNITION

### 2.1 PROBLEM FORMULATION

We consider the unsupervised phoneme recognition problem. Specifically, for a given sequence of input feature vectors $x = (x_1, \ldots, x_T)$, we want to map it into a sequence of phonemes $q = (q_1, \ldots, q_U)$, where $x_t \in \mathbb{R}^m$ is an $m$-dimensional input acoustic feature vector (e.g., mel-frequency cepstral coefficients (MFCC)), $q_i \in \mathcal{Y}$ is a categorical variable representing the phoneme class, $\mathcal{Y}$ denotes the set of phonemes, $T$ is the length of the input sequence, and $U$ is the length of the output sequence. Note that the length of the input sequence is usually much larger than that of the output sequence.[2] This is because speech data have a special segmental structure where a segment of consecutive input frames are associated with one phoneme class, as shown in Figure 1. Furthermore, the length and boundaries of each phoneme segment are generally varying and unknown a priori. We introduce a binary variable $b_t \in \{0, 1\}$ to characterize the segment boundaries: $b_t = 1$ denotes the start of a new phoneme segment (see Figure 1). Let $y_t \in \mathcal{Y}$ be the frame-wise phoneme label indicating the phoneme class that the $t$-th input frame $x_t$ belongs to. In this work, we focus on learning a framewise phoneme classifier that maps the input sequence $x_1, \ldots, x_T$ into its frame-wise label sequence $y = (y_1, \ldots, y_T)$. Once this is done, we could use a standard speech decoder to obtain the desired phoneme sequence $q_1, \ldots, q_U$ from $y_1, \ldots, y_T$. We model the framewise phoneme classifier $p_\theta(y_t|x_t)$ (i.e., the posterior probability of the frame label $y_t$ given the input $x_t$) by a context dependent DNN (Dahl et al., 2012), where $\theta$ denotes the model parameter and the input feature vector $x_t$ is a concatenation of the acoustic feature vectors within a context window around time $t$. We may also use other model architectures such as recurrent neural network (RNN), which are left as the future work. The objective of our unsupervised learning algorithm is to learn the model parameter $\theta$ from: (i) a training set of input sequences $\mathcal{D}_x$, and (ii) a pretrained phoneme language model $p_{\text{LM}}(q)$. Note that the language model could be trained from a separate (text) corpus in an unsupervised manner so that there is no supervision at any level.

There are two main challenges for unsupervised speech recognition: (i) how to learn the classifier $p_\theta(y_t|x_t)$ from $\mathcal{D}_x$ and $p_{\text{LM}}(q)$ for a given set of segmentation boundaries, and (ii) how to estimate

---

[2]We will use notation $t$ to index input frames and use notation $i$ to index segments (or phonemes).

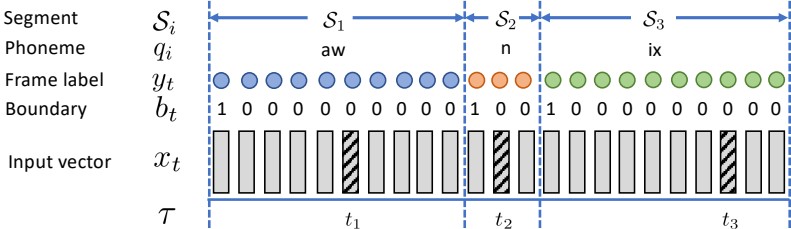

Figure 1: Segmental structure of speech data. Circles of the same color denote the same frame label in each segment. The shaded input vectors represent the sampled vectors to compute $J_{\mathrm{ODM}}(\theta)$.

the segmentation boundaries in an unsupervised manner. Unlike text where content can be broken into word units relatively easily, speech inputs are continuous and thus it is difficult to obtain the phoneme boundaries. This challenge is unique to unsupervised speech recognition and does not appear in e.g. unsupervised machine translation. In the following sections, we address the above challenges by developing a new unsupervised learning cost function by extending the Empirical-ODM cost in (Liu et al., 2017) to segmental structures. Furthermore, we develop a maximum a posteriori (MAP) estimator to refine the segmentation boundaries based on the current $p_\theta(y_t|x_t)$. Our algorithm alternates between these two steps, and after the iteration completes, we employ an unsupervised HMM training technique to further boost our unsupervised results.

## 2.2 UNSUPERVISED FRAME CLASSIFICATION WITH GIVEN SEGMENTATION BOUNDARIES

In this section, we develop an unsupervised algorithm to learn the classification model $p_\theta(y_t|x_t)$ with a given set of segmentation boundaries $\{b_t\}$. To this end, we define a new unsupervised learning cost function that exploits the segmental structure of the problem. Specifically, our new cost function is based on the following two observations: (i) the distribution of the predicted outputs across consecutive segments shall match the phoneme language model $p_{\mathrm{LM}}(q)$, and (ii) the predicted outputs within each segment should be equal to each other as they belong to the same phoneme. Accordingly, our unsupervised cost function consists of two parts, characterizing the above inter-segment and intra-segment distributions, respectively.

We first define the cost function associated with the inter-segment distribution. Before that, we introduce the following terms and notations, which are also illustrated in Figure 1. To simplify notation, we assume that all the utterances in $\mathcal{D}_x$ are concatenated into one long sequence. Let there be a total of $K$ segments in the entire training set $\mathcal{D}_x$ and let $\mathcal{S}_i$ be a set that includes all the time indexes in the $i$-th segment. We use $\tau = (t_1, \ldots, t_K)$ to denote a sequence of time indexes sampled from $\mathcal{S}_1, \ldots, \mathcal{S}_K$, one per segment, i.e., $t_i \in \mathcal{S}_i$. Without loss of generality, we consider $N$-gram phoneme language models throughout this work and define $p_{\mathrm{LM}}(z) \triangleq p_{\mathrm{LM}}(q_{i-N+1} = z_1, \ldots, q_i = z_N)$, where $z = (z_1, \ldots, z_N) \in \mathcal{Y}^N$ denotes a particular $N$-gram. Furthermore, let $\tau_i = (t_{i-N+1}, \ldots, t_i)$ be a length-$N$ contiguous subsequence of $\tau$ that ends at $t_i$. We use the compact notation $x_{\tau_i}$ and $y_{\tau_i}$ to represent $(x_{t_{i-N+1}}, \ldots, x_{t_i})$ and $(y_{t_{i-N+1}}, \ldots, y_{t_i})$, respectively. Then, the cost function that characterizes the inter-segment output distribution match is defined as:

$$J_{\mathrm{ODM}}(\theta) = - \sum_{\tau \in \mathcal{S}_1 \times \cdots \times \mathcal{S}_K} \sum_{z \in \mathcal{Y}^N} p_{\mathrm{LM}}(z) \ln \overline{p}_\theta^\tau(z) \tag{1}$$

where $\overline{p}_\theta^\tau(z) = \frac{1}{K} \sum_{i=1}^{K} p_\theta(y_{\tau_i} = z|x_{\tau_i})$ is defined as the inter-segment output distribution with $p_\theta(y_{\tau_i} = z|x_{\tau_i}) \triangleq \prod_{j=i-N+1}^{i} p_\theta(y_{t_j} = z_j|x_{t_j})$. The cost function (1) generalizes the Empirical-ODM cost in Liu et al. (2017) to the segmental structures, and it degenerates into the original Empirical-ODM cost when there is only one frame in each segment. The probability $\overline{p}_\theta^\tau(z)$ characterizes the empirical $N$-gram frequency of the predicted output across $N$ consecutive segments, and the cost function measures the cross-entropy between the pretrained $N$-gram LM $p_{\mathrm{LM}}(z)$ and $\overline{p}_\theta^\tau(z)$. This form of cost function enjoys several properties that are suitable for unsupervised learning of sequence classifiers, and the readers are referred to Liu et al. (2017) for more detailed discussions.

Next, we define the cost function that characterizes the intra-segment distribution matching as:

$$J_{\text{FS}}(\theta) = \sum_{i=1}^{K} \sum_{t,t+1 \in \mathcal{S}_i} \sum_{y \in \mathcal{Y}} \Big( p_\theta(y_t = y | x_t) - p_\theta(y_{t+1} = y | x_{t+1}) \Big)^2 \tag{2}$$

where the subscript "FS" stands for frame-wise smoothness. The cost (2) encourages the predictions for adjacent frames within the same segment to be similar. It captures the strong intra-segment temporal structure in speech signals that complements the cost (1). Our final unsupervised cost function combines the inter-segment and intra-segment distribution matching via:

$$\min_\theta \big\{ J(\theta) \triangleq J_{\text{ODM}}(\theta) + \lambda J_{\text{FS}}(\theta) \big\} \tag{3}$$

where $\lambda$ is a parameter controlling the trade-off between the two parts. We call the cost function $J(\theta)$ Segmental Empirical-ODM as it captures the segmental structure through the inter-segment and intra-segment terms. To optimize this cost function, we sample a sequence $\tau$ at the beginning of each epoch and applies stochastic gradient descent (SGD) with momentum to update $\theta$. Note that in $J_{\text{ODM}}(\theta)$, there is an empirical average over all $K$ segments in $\overline{p}_\theta^\tau(z)$, which is inside the logarithmic function. This makes stochastic gradient descent intrinsically biased if we also sample this empirical average by a mini-batch average. To alleviate this effect, we use a large mini-batch size to estimate the stochastic gradients.

Note that our method directly optimizes the classifier $p_\theta(y_t|x_t)$ that takes the raw acoustic feature vector $x_t$ (e.g., MFCC features) and maps it into output space. This is different from the previous work (Liu et al., 2018), which first performs clustering in the speech space and then maps the clusters into output space using adversarial training. This makes its performance upper-bounded by the purity of the initial clusters since input frames of different phonemes may be mapped into the same cluster. In contrast, our algorithm is end-to-end trained without using a separate clustering algorithm. This enables us to outperform the cluster purity upper bound, as shown in our experiment section.

## 2.3 Segmentation boundary refinement using the classification model

In this section, we develop an approach to refining the estimated segmentation boundaries $\{b_t\}$ using a learned framewise phoneme classifier $p_\theta(y_t|x_t)$. More formally, for each input utterance sequence $x = (x_1, \ldots, x_T)$, we would like to infer the corresponding boundary sequence $b = (b_1, \ldots, b_T)$. We propose a simple yet effective MAP estimation strategy by recognizing the fact that $b_t = \mathbb{I}(y_t \neq y_{t-1})$. Therefore, we can perform an MAP estimate for $y = (y_1, \ldots, y_T)$ and then predict the boundaries by $b_t = \mathbb{I}(y_t \neq y_{t-1})$. The MAP estimator of $y$ can be expressed as (see Appendix A):

$$\arg \max_y p(y|x) = \arg \max_y \prod_{t=1}^{T} p(y_t | y_1, \ldots, y_{t-1}) \frac{p_\theta(y_t | x_t)}{p(y_t)} \tag{4}$$

Note that $p(y_t|y_1, \ldots, y_{t-1})$ is the transition probability of the frame labels. Assuming that $y_t$ belongs to the $i$-th segment, we can express $p(y_t|y_1, \ldots, y_{t-1})$ as:

$$p(y_t|y_1, \ldots, y_{t-1}) = \mathbb{I}(y_t = y_{t-1}) p(y_t = y_{t-1}) + \mathbb{I}(y_t \neq y_{t-1}) p(y_t \neq y_{t-1}) p_{\text{LM}}(q_i = y_t | q_{1:i-1})$$
$$= \mathbb{I}(y_t = y_{t-1}) p(b_t = 0) + \mathbb{I}(y_t \neq y_{t-1}) p(b_t = 1) p_{\text{LM}}(q_i = y_t | q_{1:i-1}) \tag{5}$$

where $q_{1:i-1}$ denotes the previous $i - 1$ phonemes that the sequence $y_1, \ldots, y_{t-1}$ has traversed. The first term in (5) characterizes the probability that $y_t$ stays in the same phoneme segment as $y_{t-1}$ and the second term defines the probability that $y_t$ belongs to a new phoneme segment. Note that $p_{\text{LM}}(q_i = y_t | q_{1:i-1})$ in (5) could be obtained from the phoneme language model. It remains to estimate $p(b_t = 0)$ and $p(b_t = 1)$, which we approximate by $p(b_t = 0|x)$ and $p(b_t = 1|x)$, respectively. To obtain $p(b_t = 1|x)$, we leverage the work of Wang et al. (2017), which shows that the temporal structure of the gate signals in a gated RNN (GRNN) auto-encoder is highly correlated with phoneme boundaries. Therefore, we apply a sigmoid function to the Difference GAS value (defined as Wang et al. (2017)) to obtain $p(b_t = 1|x)$. With all these information, we substitute (5) into (4) and perform a beam search to solve (4) for an approximate MAP estimate of $y$. It follows that $b_t = \mathbb{I}(y_t \neq y_{t-1})$ and we have refined the boundaries using $p_\theta(y_t|x_t)$.

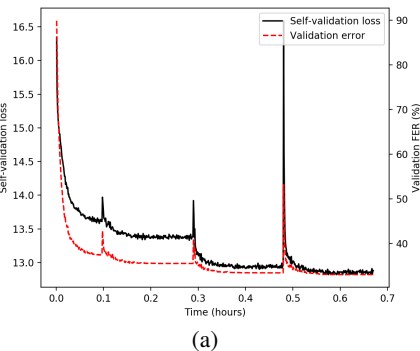 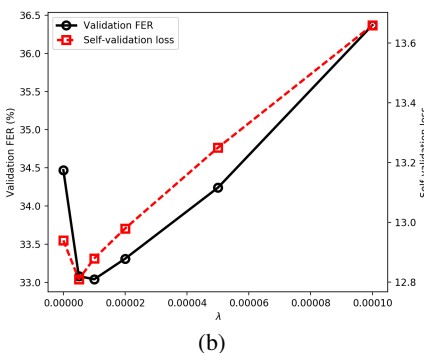

(a)                                                    (b)

Figure 2: Self-validation metric. (a) The learning curves of the self-validation loss and the validation FER. (b) The self-validation loss and the validation FER for different values of $\lambda$ in (3).

## 2.4 ALTERNATING TRAINING ALGORITHM

Our unsupervised learning algorithm for $p_\theta(y_t|x_t)$ alternates between the above two steps of estimating $\theta$ for a given $b$ and refining $b$ for a given $\theta$. The overall algorithm is summarized in Algorithm 1. We initialize the algorithm by thresholding the temporal update gate activation in Wang et al. (2017) to obtain an initial rough estimate for $b$. After the training converges,[3] we could apply the unsupervised HMM training technique discussed in Section 3 to further boost the performance. Note that although the training process requires boundary estimation, at testing stage, it is not necessary because the learned $p_\theta(y_t|x_t)$ could be used in standard speech decoders just as supervised models.

---

**Algorithm 1:** Training Algorithm

---

**Input:** Phoneme language model $p_{\text{LM}}(z)$, Training data $\mathcal{D}_x$, initial boundary $b_{\text{init}}$ obtained by using techniques proposed in Wang et al. (2017).
**Output:** Model parameter $\theta$
1 Initialization for parameters $\theta$.
2 **while** *not converged* **do**
3     Given a set of boundaries $b$, obtain a new $\theta$ by optimizing (3).
4     Given the model parameter $\theta$, obtain a new estimate for the boundaries $b$ by optimizing (4).
5 **end**

---

## 2.5 UNSUPERVISED MODEL SELECTION

Since there are no labeled data during the training process, we need to develop an unsupervised self-validation metric to perform model selection. We propose to use the value of the loss function (1) on a heldout validation set (including only input features) to perform model selection. This self-validation loss gives us an estimate of which model configuration is better, and is used to a) determine when to stop training, and b) to select the best hyper-parameters. To validate the effectiveness for our self-validation loss, we show the learning curves of this self-validation loss and the validation frame error rate in Figure 2(a). We observe that the self-validation loss aligns well with the true validation error. Furthermore, in Figure 2(b) we plot the self-validation loss and the validation FER for different values of $\lambda$, which shows that the two metrics are highly correlated. The results demonstrate that the self-validation loss can be effectively used to select a good model.

## 3 UNSUPERVISED HMM TRAINING

To further improve the performance of proposed unsupervised speech recognition system, we explore the semi-supervised hidden Markov model (HMM) training strategy (Zavaliagkos et al., 1998; Kemp and Waibel, 1999) that has commonly been used in speech recognition. The semi-supervised HMM training is an effective technique where a seed model trained on a relatively small amount

---

[3] We observe in our experiments that two iterations are sufficient to converge.

of labeled speech data is used for providing labels for larger amount of non-transcribed speech data for iterative model refinement. A major difference of the HMM training strategy used in this work compared to the ones used in semi-supervised learning is that we use the transcripts generated from proposed unsupervised speech recognition system (i.e., predicted labels for 3969 TIMIT training utterances) for bootstrapping the training of HMM-based models. Therefore, the training of HMM models in this work does not require any manually provided supervised information. The training of HMM based speech recognition models follows the standard recipes in Kaldi speech recognition toolkit (Povey et al., 2011). We experimented with monophone and triphone models with MFCC feature as input as well as more advanced speaker adaptive training (SAT) (Matsoukas et al., 1997) approach with feature space maximum likelihood linear regression (fMLLR) (Gales et al.) as input.

## 4 EXPERIMENTS

### 4.1 EXPERIMENT SETTING

We perform experiments on the TIMIT dataset where 6300 prompted English speech sentences are recorded. The preparation of training and test sets follow the standard protocol of the TIMIT dataset. The phoneme transcription of these utterances are manually segmented and labelled with a lexicon of 61 distinct phoneme classes. These phoneme labels are mapped to 39 phoneme classes for scoring phone error rate (Lee and Hon, 1989). We use 39 dimensional feature vectors including 13 mel-frequency cepstral coefficients (MFCC) plus its acceleration features that are extracted with 25 ms Hamming window at 10 ms interval. The classifier $p_\theta(y_t|x_t)$ is modeled by a fully connected neural network with one hidden layer of $512$ ReLU units. The input to the neural network is a concatenation of frames within a context window of size $11$. We follow the default hyper-parameters in Wang et al. (2017) to estimate the phoneme boundaries, which are used to initialize our algorithm. The optimization of (3) is performed with momentum SGD with a fixed schedule of increasing batch size from 5000 to 20000. $\lambda$ in (3) is chosen to be $10^{-5}$. We use both frame error rate (FER) and phone error rate (PER) as our evaluation metrics. Details of the experiment setting and other hyper-parameters can be found in Appendix B.

### 4.2 BASELINE METHODS

**Adversarial Mapping**   The first baseline we consider is the work by Liu et al. (2018), which learns an unsupervised embedding by a sequence-to-sequence autoencoder followed by k-means clustering. Each cluster is then mapped to a phoneme by adversarial training between the cluster sequences and the phoneme sequence. The phoneme boundaries are given by a supervised oracle.

**Cluster Purity**   The accuracy of Adversarial Mapping (Liu et al., 2018) is upper-bounded by the cluster purity, which is the frame accuracy when assigning all the frames in each cluster to its most frequent phonemes. It is a supervised baseline since it relies on the phoneme labels. We show the cluster purity for 1000 clusters, which is the largest number of clusters used by Liu et al. (2018).

**Supervised Neural Network**   We train a supervised neural network with the same architecture as our unsupervised model with standard cross-entropy loss.

**Supervised RNN Transducer**   It is one of the state-of-the-art methods, which learns a BiLSTM-RNN Transducer with supervised learning (Graves et al., 2013).

### 4.3 EXPERIMENT RESULTS

**Unsupervised speech recognition with oracle boundary**   In our proposed unsupervised learning algorithm, we use the cost function (3) to train the classifier, which is different from the cross entropy cost in supervised learning. To examine the effect of replacing cross entropy with this new unsupervised cost function, we first conduct experiments under the oracle phoneme boundaries. This setting also allows us to compare our method to the one in Liu et al. (2018), which assumes oracle phoneme boundaries. Specifically, we consider two settings. In the first setting, we follow the standard TIMIT partition to divide the training data into a training and validation sets of 3696 and 400 utterances, respectively. We use the phoneme transcription of the 3696 utterances to train our language model $p_{\mathrm{LM}}(z)$ and call this setting "matching language model". Then we use this learned $p_{\mathrm{LM}}(z)$ together with the 3696 input utterances to train our model by minimizing (3). In the second setting, we divide

| Language Model | Matcthing | | | Non-Matcthing | |
|---|---|---|---|---|---|
| Evaluation Metric | FER* | FER | PER | FER | PER |
| Supervised Methods | | | | | |
| RNN Transducer (Graves et al., 2013) | – | – | 17.7 | – | – |
| Supervised Neural Network | 35.5 | 31.0 | 30.2 | 31.7 | 31.1 |
| Cluster Purity (1000) (Liu et al., 2018) | 41.0 | – | – | – | – |
| Unsupervised Methods | | | | | |
| Adversarial Mapping (Liu et al., 2018) | 47.5 | – | – | – | – |
| Our Model | 38.2 | 33.3 | 32.5 | 40.0 | 40.1 |

Table 1: Phoneme classification results when phoneme boundaries are given by a supervised oracle.

| Language Model | Matcthing | | Non-Matcthing | |
|---|---|---|---|---|
| Evaluation Metric | FER | PER | FER | PER |
| Supervised Methods | | | | |
| RNN Transducer (Graves et al., 2013) | – | 17.7 | – | – |
| Supervised Neural Network | 31.0 | 30.2 | 31.7 | 31.1 |
| Unsupervised Methods | | | | |
| Our Model: 1st iteration | 47.4 | 47.0 | 63.5 | 61.7 |
| Our Model: 2nd iteration | 45.4 | 42.6 | 51.6 | 49.1 |
| Our Model: 2nd iteration + HMM (mono) | – | 41.5 | – | 44.7 |
| Our Model: 2nd iteration + HMM (tri) | – | 39.4 | – | 44.9 |
| Our Model: 2nd iteration + HMM (tri + SAT) | – | 36.5 | – | 41.6 |

Table 2: Results for fully unsupervised phoneme classification.

the data into a training and a validation sets of 3000 and 1096 utterances, respectively. We train our language model $p_{LM}(z)$ on the phoneme transcription of the 1096 utterances, and use the learned $p_{LM}(z)$ with the other 3000 input utterances to train our model by minimizing (3). In this setting, the training corpus for $p_{LM}(z)$ does not overlap with the 3000 input training utterances, and we call it "non-matching language model". Note that both settings are unsupervised since we do not use any phoneme label in our training process. The only difference is the source of the language model. The results of our algorithm and other baselines are summarized in Table 1. Although we are still far away from the state-of-the-art, in the matching LM setting, the performance of our algorithm (32.5% PER) is approaching that of the supervised system with the same model architecture (30.2% PER). This is an encouraging result, showing that replacing the supervised cross entropy loss with our unsupervised learning cost does not degrade the performance much. On the other hand, when we use non-matching LM, the gap becomes larger (40.1% vs 31.1% in PER). We think the reason is due to the discrepancy of the output distributions between the two sets and the reduced training corpus for $p_{LM}(z)$. We believe such a discrepancy could be alleviated by using a large-scale dataset. Other than the standard FER and PER, we additionally show the evaluation result for FER* where the starting and ending silences are removed following the setting in Liu et al. (2018). We observe that our approach significantly outperforms the unsupervised Adversarial Mapping method in Liu et al. (2018), and even outperforms the Cluster Purity (supervised upper bound in Liu et al. (2018)). This result is not surprising since the clustering does not exploit the output distribution, and may group inputs of different phonemes into the same cluster.

**Fully unsupervised speech recognition** We now consider the fully unsupervised setting where only input speech features and a language model is given. The phoneme boundaries are not given and has to be estimated in an unsupervised manner using our Algorithm 1. We show the quality of the learned model after each iteration of the learning process in Table 2. And we observe that our iteration process improves the results by a great margin especially in the non-matching LM case, significantly lowering the FER and PER by over 10%. This demonstrates that our boundary refining process has resulted in a better set of boundaries, which greatly improves the output distribution

| Evaluation Metric | Recall | Precision | F-score | R-value |
|---|---|---|---|---|
| Dusan and Rabiner (2006) | 75.2 | 66.8 | 70.8 | – |
| Qiao et al. (2008) | 77.5 | 76.3 | 76.9 | – |
| Lee and Glass (2012) | 76.2 | 76.4 | 76.3 | – |
| Rasanen (2014) | 74.0 | 70.0 | 73.0 | 76.0 |
| Hoang and Wang (2015) | – | – | 78.2 | 81.1 |
| Michel et al. (2016) | 74.8 | 81.9 | 78.2 | 80.1 |
| Wang et al. (2017) | 78.2 | 82.2 | 80.1 | 82.6 |
| Ours refined boundaries | **80.9** | **84.3** | **82.6** | **84.8** |

Table 3: Results for unsupervised phoneme boundary segmentation.

matching. Moreover, we also report the results of using unsupervised HMM training where the PER can be further improved. In the matching LM setting, HMM training with monophone, triphone, and speaker adaptation training (SAT) improves the PER by a similar amount. In the non-matching LM setting, HMM training significantly improves the PER, and SAT additionally improves 3% in PER. Overall, our hybrid system with matching and non-matching LM achieved 36.5% and 41.6% PER, respectively, which is only 10% below the supervised system of the same architecture.

**Unsupervised Phoneme Segmentation**    To understand how much our proposed boundary refinement method in Section 2.3 improves the segmentation quality, we follow the setting in previous works and report in Table 3 the recall, precision, F-score, and R-value with a 20-ms tolerance window on TIMIT's training set (Scharenborg et al., 2010; Versteegh et al., 2016; Rasanen, 2014). We compare our results (obtained with matching LM) with several unsupervised phoneme segmentation methods (Dusan and Rabiner, 2006; Qiao et al., 2008; Lee and Glass, 2012; Rasanen, 2014; Hoang and Wang, 2015; Michel et al., 2016; Wang et al., 2017). Note that our refined segmentation significantly improves over the initial boundaries generated by Wang et al. (2017) and also outperforms other baselines. This result also confirms its contribution to the much improved phoneme recognition performance in the 2nd iteration (see "Our Model: 2nd iteration" in Table 2) to the 1st iteration. However, we emphasize that our method is designed towards unsupervised speech recognition rather than unsupervised phoneme segmentation. Estimating the segmentation boundary only serves as an auxiliary task to enable the unsupervised learning of the recognition model. And in the testing stage, there is no need to estimate the segmentation boundaries. Instead, our trained model could be directly used with a speech decoder just as any supervised recognition model would do.

**Further analysis**    We include some further experiments and analysis in Appendix C, where we show the importance of the frame smoothness term in (3). We also compare the performance of our unsupervised algorithm to supervised learning with different amounts of labeled training data.

## 5  RELATED WORK

**Unsupervised sequence-to-sequence learning**    Recently, unsupervised sequence-to-sequence learning has achieved great success in several problems. Liu et al. (2017) showed that it is possible to learn a sequence classifier without any labeled data by exploiting the output sequential structure using an unsupervised cost function named Empirical-ODM. Artetxe et al. (2018) and Lample et al. (2018) showed that unsupervised neural machine translation (uNMT) systems can be achieved by utilizing cross-lingual alignments and an adversarial structure without any form of parallel information. The success in the unsupervised sequence-to-sequence learning in various applications shed light on building our fully unsupervised speech recognition system. In particular, our work extends the Empirical-ODM in Liu et al. (2017) to problem with segmental structures.

**Unsupervised speech segmentation**    One line of unsupervised segmentation methods designs robust acoustic features that are likely to remain stable within a phoneme, and capture the change of

features for phoneme boundaries (Esposito and Aversano, 2005; Hoang and Wang, 2015; Khanagha et al., 2014; Rasanen et al., 2011; Michel et al., 2016; Wang et al., 2017). Another line of research uses a simpler segmentation method as an initialization, and jointly trains the segmenting and acoustic models for phonemes or words (Kamper et al., 2015; Glass, 2003; Siu et al., 2014; Lee and Glass, 2012). Qiao et al. (2008) use dynamic programming methods in order to the derive optimal segmentation, but requires the number of segments and is not fully unsupervised. In Wang et al. (2017), the authors use the update gate of a GRNN autoencoder to discover the phoneme boundaries.

**Unsupervised spoken term discovery**   Recently, the discovery of acoustic tokens including subword and word units has become a popular research topic (Dunbar et al. (2017); Versteegh et al. (2016); Burget et al.). The term "Spoken term discovery" includes lexicon discovery, word segmentation, and subword matching (Dunbar et al. (2017)). The standard approaches segment audio signals that are acoustically similar, and cluster the obtained segmented signals (Lee and Glass, 2012; Glass, 2012; Park and Glass, 2008; Driesen et al., 2012). Walter et al. (2013) uses the discovered unit index sequence as the transcription for the acoustic model training, similar to the HMM training in section 3. Kamper et al. (2017) iterates between the clustering and segmentation steps. Ondel et al. (2016) improves upon previous methods by replacing Gibbs sampling by variational inference, and Ondel et al. (2017) further improves the result by including a bigram language model. The effectiveness of these approaches has been demonstrated on query-by-example spoken term detection or by calculating the normalized mutual information between the self-discovered units and the actual labels. Overall, these methods differ from our method in that they segment and cluster the raw speech signals to self-discovered units, but does not recognize them into phoneme or word labels directly. More recently, Chung et al. (2018) show that unsupervised spoken word classification is possible by using adversarial cross-modal alignments similar to that in uNMT systems.

**Unsupervised speech recognition with oracle segmentation**   There have been several attempts (Liu et al., 2018; Chen et al., 2018) on building an unsupervised speech recognition model inspired by the success of the uNMT. These methods first learn an embedding from the acoustic data, and then map the clustered embeddings to the output space by either adversarial training or iterative mapping. In contrast, our approach learns a neural network model that directly maps the raw acoustic features into the output space by optimizing the Segmental Empirical-ODM cost, and outperforms the upper bound of the above cluster-based approaches. Furthermore, all methods in Liu et al. (2018); Chen et al. (2018) assume that the phoneme boundaries are given by a supervised oracle. In contrast, our method iteratively estimates the boundaries without any labeled data, making it fully unsupervised.

## 6   CONCLUSION

We have developed a fully unsupervised learning algorithm for phoneme recognition. The algorithm alternates between two steps: (i) learn a phoneme classifier for a given set of phoneme segmentation boundaries, and (ii) refining the phoneme boundaries based on a given classifier. For the first step, we developed a novel unsupervised cost function named Segmental Empirical-ODM by generalizing the work (Liu et al., 2017) to segmental structures. For the second step, we developed an approximate MAP approach to refining the boundaries obtained from Wang et al. (2017). Our experimental results on TIMIT phoneme recognition task demonstrate the success of a fully unsupervised phoneme recognition system. Although the fully unsupervised system is still far away from the state-of-the-art supervised methods (e.g., supervised RNN transducer), we show that with oracle boundaries the performance of our algorithm could approach that of the supervised system with the same model architecture. This demonstrates the potential of our method if, in future work, we can further improve the accuracy of boundary estimation. We want to further point out that the techniques we proposed in this paper, although was evaluated in speech recognition, can be exploited to attack other similar sequence recognition problems where the source and destination sequences have different lengths and labels are not available or hard to get.

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

# SUPPLEMENTARY MATERIAL

## A   DERIVATION OF THE MAP ESTIMATE FOR THE SEGMENTATION BOUNDARIES

In this appendix, we derive the MAP estimate for $y = (y_1, \ldots, y_T)$ given an input utterance sequence $x = (y_1, \ldots, y_T)$. Specifically, we have

$$
\begin{aligned}
\arg \max_y p(y|x) &= \arg \max_y p(y, x) \\
&= \arg \max_y p(y) p(x|y) \\
&\overset{(a)}{=} \arg \max_y \prod_{t=1}^{T} p(y_t|y_1, \ldots, y_{t-1}) p_\theta(x_t|y_t) \\
&= \arg \max_y \prod_{t=1}^{T} p(y_t|y_1, \ldots, y_{t-1}) \frac{p_\theta(y_t|x_t) p(x_t)}{p(y_t)} \\
&\overset{(b)}{=} \arg \max_y \prod_{t=1}^{T} p(y_t|y_1, \ldots, y_{t-1}) \frac{p_\theta(y_t|x_t)}{p(y_t)}
\end{aligned}
\tag{6}
$$

where in step (a) we approximate the $p(x|y)$ by its factored form and in step (b) we dropped the constant term that is independent of $y$.

## B   DETAILED EXPERIMENT SETTING

We perform experiments on the TIMIT dataset where 6300 prompted English speech utterances are recorded. The phoneme transcription of these utterances are manually segmented and labelled with a lexicon of 61 distinct phoneme classes, where we compact the 61 phoneme classes into 48 phone classes and train the language model with the validation dataset, which is later used to train our main algorithm. These 48 phoneme classes are mapped to 39 phoneme classes for scoring phone error rate (Lee and Hon, 1989).

The 39 dimensional feature vectors including 13 mel-frequency cepstral coefficients (MFCC) plus its acceleration features that are extracted with 25 ms Hamming window at 10 ms interval. The classifier $P_\theta$ is modeled by a fully connected neural network with one hidden layer of 512 ReLU units. The input to the neural network is a concatenation of frames within a context window of size 11, and we repeat the starting or ending frames if the window has reached the start or end of the sentence.

The optimization of (3) is performed with momentum SGD with momentum 0.9 and learning rate of $10^{-3}$ with learning rate decay with a fixed schedule of increasing batchsize from 5000 to 20000 and decreasing temperature for softmax. The scheduler parameter is listed below: first 200 epochs with batchsize 500 and temperature 1.0, next 300 epochs with batchsize 5000 and temperature 0.9, followed by 300 epochs with batchsize 10000 and temperature 0.8, finally 300 epochs with batchsize 20000 and temperature 0.7. Whenever the batchsize is increased, we set the learning rate to the inital learning rate value. The scheduling procedure is determined by self-validation, and is not extensively tuned during the experiments.

In our experiments, we chose $N = 5$ for the N-gram in (1), and for computational issues we only consider the most frequent 10000 5-grams. We do not observe any noticeable performance drop by considering only the 10000 5-grams. Among the 5-gram language model $P_{\text{LM}}$, 69553 5-grams (out of $48^5$ possible 5-grams) are non-zero, and the top 10000 5-grams account for almost half of the probability. To sample $\tau$, we use a standard truncated normal distribution for sampling the frame in every segment, with some necessary scaling and rounding. The truncated distribution is to ensure that our sampling will give us bounded frames that lie in the correct segment. This distribution can also be replaced by the uniform distribution. To obtain $p(b_t = 1|x)$, we apply a sigmoid function to the normalized Difference GAS value (defined as Wang et al. (2017)) that are local maxima. For the Difference GAS value that are not local maxima, we set it to -0.1 to get a low probability.

We randomly sample 10000 continuous frames to optimize (2), which is sampled every batch. $\lambda$ in (3) is chosen to be $10^{-5}$. We use both frame error rate (FER) and phone error rate (PER) as our evaluation metrics. All phone error rate (PER) results reported has been obtained by a Kaldi decoder by considering the per-frame softmax value and the language model, and the weight between the two set to 1, which is fixed in all unsupervised setting.

## C  ADDITIONAL EXPERIMENTS AND ANALYSIS

First, we examine the importance of the frame smoothness term in (3) in the fully unsupervised setting. In Figure 3(a), we show the FER of our model after the first iteration of Algorithm 1 for different values of $\lambda$. Note that when $\lambda$ is close to the order of $10^{-5}$, the result does not differ a lot from the best result. However, when $\lambda$ is set to zero, the performance degrades significantly. This confirms the importance of incorporating the temporal structure of speech data into the cost function, as discussed in Section 2.2. Second, we would like to study another important question regarding our unsupervised learning method: how much labeled data is it equivalent to? In Figure 3(b), we show the supervised neural network with different sizes of training data, where x-axis is the percentage of the original labeled set being used to train the model. We observe that with oracle boundary and matching LM, our algorithm is equivalent to supervised learning with $30\%$ labeled data. With unsupervised boundary estimation, we still see a big performance loss. Therefore, it is critical to improve the boundary estimation performance in our future work.

| $\lambda$ | FER |
|:---:|:---:|
| $5 \times 10^{-6}$ | 49.2 |
| $1 \times 10^{-5}$ | 47.4 |
| $2 \times 10^{-5}$ | 48.1 |
| 0 | 70.2 |

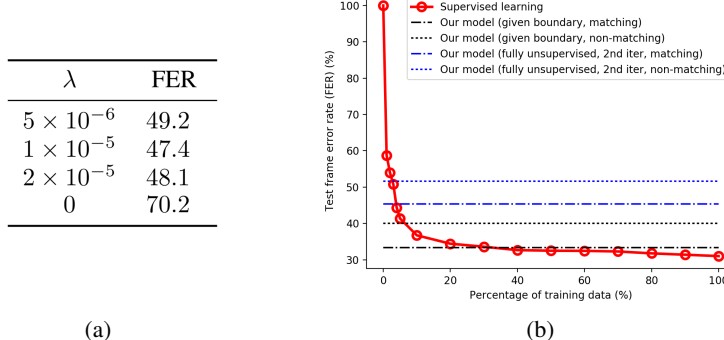

(a)                                         (b)

Figure 3: Further analysis of our algorithm. (a) Influence of $\lambda$ after the 1st iteration of fully unsupervised learning with matching language model. (b) Equivalent amount of labeled data.

