# OpenReview forum: "Unsupervised Speech Recognition via Segmental Empirical Output Distribution Matching"
_ICLR.cc/2019/Conference_

### Official Review · AnonReviewer1 · 2018-11-01
**Interesting approach, but should improve**

**Rating:** 7
**Confidence:** 4

**Review:**

Overview:

This paper proposes a new approach to do unsupervised phoneme recognition by learning from unlabelled speech in combination with a trained phoneme language model. The proposed loss function is a combination of a term encouraging the language model of predicted phonemes to match the given language model distribution, and a term to encourage adjacent speech frames to be assigned to the same phoneme class. Phoneme boundaries are iteratively refined using a separate model. Experiments where a hidden Markov model is applied on top of the predicted phonemes are also performed.


Main strengths:

The paper is clear and addresses a very important research problem. The approach and losses proposed in Section 2 have also not been proposed before, and given that an external language model is available, are very natural choices.


Main weaknesses:

The main weakness of this paper is that it does not situate itself within the rich body of literature on this problem.  I give several references below, but I think the authors can include even more studies -- there are several studies around "zero-resource" speech processing, and I would encourage the authors to work through the review papers [1, 6].

Concretely, I do not think the authors can claim that "this is the first fully unsupervised speech recognition method that does not use any oracle segmentation or labels." I think it could be argued that the system of [3] is doing this, and there are even earlier studies. I also don't think this claim is actually necessary since the paper has enough merit to stand on its own, as long as the related work is discussed properly.

For instance, the proposed approach shares commonalities with several other approaches: [2] also used two separate steps for acoustic modelling and boundary segmentation; [4, 7, 8] builds towards the setting where non-matching text data is available (for language model training) together with untranscribed speech for model development; the approach of [5] uses a very similar refinement step to the one described in Section 3, where an HMM model is initialised and retrained on noisy predicted labels.

In the experiments (Section 4), it would also be useful to report more fine-grained metrics. [6] gives an overview of several of the standard metrics used in this area, but at a minimum phoneme boundary recall, precision and F-scores should be reported in order to allow comparisons to other studies.


Overall feedback:

Given that this paper is situated within the broader context of this research area, which already has a small community around it, I think the novelty in the approach is strong enough to warrant publication given that the additional metrics are reported in the experiments.


Papers/links that should be reviewed and cited:

1. E. Dunbar et al., "The Zero Resource Speech Challenge 2017," in Proc. ASRU, 2017.
2. H. Kamper, K. Livescu, and S. Goldwater. An embedded segmental k-means model for unsupervised segmentation and clustering of speech. in Proc. ASRU, 2017.
3. Lee, C.-y. and Glass, J. R. A nonparametric Bayesian approach to acoustic model discovery. ACL, 2012.
4. Ondel, Lucas, Lukaš Burget, Jan Černocký, and Santosh Kesiraju. "Bayesian phonotactic language model for acoustic unit discovery." In Acoustics, Speech and Signal Processing (ICASSP), 2017 IEEE International Conference on, pp. 5750-5754. IEEE, 2017.
5. Walter, O., Korthals, T., Haeb-Umbach, R., and Raj, B. (2013). A hierarchical system for word discovery exploiting DTW-based initialization. ASRU, 2013.
6. M. Versteegh, X. Anguera, A. Jansen, and E. Dupoux, "The Zero Resource Speech Challenge 2015: Proposed approaches and results,” in Proc. SLTU, 2016.
7. https://www.clsp.jhu.edu/wp-content/uploads/sites/75/2018/05/jsalt2016-burget-building-speech-recognition.pdf
8. https://www.clsp.jhu.edu/workshops/16-workshop/building-speech-recognition-system-from-untranscribed-data/

---

> ### Author Response · Authors · 2018-11-16
> **Thank you for your constructive feedback and comments**
>
> We thank the reviewer for the constructive feedback!
>
> [Related works] Thanks for the suggestion, we will adjust the claim in our revised paper. We have also incorporated these related works and discuss them thoroughly in our related work section. However, the mentioned works (including [3]) have a different focus compared to our work on unsupervised speech recognition. Specifically, they are focused on unsupervised acoustic unit discovery (AUD), i.e., finding the segmentation boundaries for the acoustic units (e.g., word and subword)  and clustering the discovered units. They did not classify acoustic inputs into phoneme or word labels in an unsupervised manner during the inference stage. In contrast, we are interested in directly learning a speech recognition model in an unsupervised manner without an intermediate clustering step; that is, our learned model will directly recognize acoustic features into phoneme labels. The estimation of the segmentation boundaries in our work is to help the training of the recognition model. In fact, although we use the segmentation boundaries generated by (Wang et al. 2017) as our initial boundaries, we could also potentially use other acoustic unit discovery methods suggested by the reviewer to initialize our algorithm, just as (Wang et al. 2017). Regarding the work in [2], although it also iterates between acoustic modeling and boundary segmentation, their acoustic modeling is mainly for cluster assignment while our work directly learns the phoneme recognition model.
>
> [Additional evaluation metrics] As suggested by the reviewer, we evaluate the precision, recall, F-score and R-value of our refined boundaries and compare them to the baselines (including Wang et al. 2017). Please refer to Table 3 of our revised paper for the full results, where our method outperforms all the other baselines by a significant margin. Below, we list the results of our method and the results from Wang et al. 2017. We can see that by refining the initial boundaries provided by (Wang et al. 2017), we improve the quality of the segmentation boundaries, which is consistent with the improved phoneme error rate (PER) in Table 2.
>
> 		        Initial boundary (Wang et al.)		Our refined boundaries
> R-value	        82.6					                        84.8
> F-score	        80.1					                        82.6
> Recall		78.2					                        80.9
> Precision	82.3					                        84.3
>
> However, we emphasize that our method is designed towards unsupervised speech recognition rather than unsupervised phoneme segmentation. Estimating the segmentation boundary only serves as an auxiliary subtask to help the training of the recognition model in an unsupervised manner. And in the testing stage, we do not estimate the segmentation boundaries for the test data. Instead, our trained model could be directly used with decoder just as any supervised recognition model would do. For this reason, the most important evaluation metric for our method is the phoneme error rate (PER). Nevertheless, the above boundary evaluation results do confirm that the improved segmentation quality indeed leads to better phoneme recognition performance (see Table 2), which also demonstrates the effectiveness of our iterative algorithm.

---

> > ### Comment · AnonReviewer1 · 2018-11-19
> > **Thank you for addressing concerns**
> >
> > I want to thank the authors for addressing my concerns.  I understand that their focus was not exactly the same as in previous work, but want to thank the authors for nevertheless adding the additional motivations and extra analysis.  I believe that this will help situate this work better within this area, and also allow better comparison with other studies.
> >
> > I have changed my overall rating from a 6 to a 7.

---

### Official Review · AnonReviewer3 · 2018-11-05
**Success on practical task using unsupervised learning, interesting problem, comprehensive study and experiments.**

**Rating:** 7
**Confidence:** 4

**Review:**

This paper proposes fully unsupervised learning algorithm for speech recognition. It involves two alternating trained component, a phoneme classifier, and a boundary refining model. The experiment results demonstrate that it achieves first success on speech recognition that approaches the supervised learning performance.

Pros:
+ The paper propose to use a frame-wise smoothing term J_FS added on J_ODM cost. In the new cost function, J_ODM controls the coarse-grained inter-segment distribution using a prepared language model P_LM, while J_FS controls the fine-grained intra-segment distribution. It is actually benefit to take use of this hierarchical 2-level scopes than only 1-level scope on evaluate the distribution mismatch in the cost function. Because otherwise, if only focus on fine-grained frame level,  much larger number of frame labels and longer N-gram have to be considered to evaluate the distribution of phoneme. Consequently, the computation can be exploding.
+ The proposed unsupervised phoneme classification method is superior to the baseline (Liu et al., 2018) because the baseline relies on a clustering which is upper-bounded by cluster purity. Directly optimize on \theta using an end-to-end scheme is preferred.
+ I like the idea to use an iterative training algorithm to jointly improve classifier parameter \theta and segment boundaries b.
+ It is quite impressive that unsupervised learning system get close to performance of supervised system on speech recognition. The proposed system also outperforms state-of-the-art baseline with large margin.
+ The settings of experiments are rather comprehensive. Especially the “non-matching language model”, tests the case where language model cannot directly estimated from training set.

Questions:
1.	In Appendix B you mentioned that for the N-gram you choose N=5. So the original language model P_LM can be a high-dim matrix with exactly 39^5 elements. How sparse is the original P_LM? It describes that 10000 elements are chosen, which are only 0.001%(=10000/39^5) of elements in the original one. How representative are they?

2.	I notice for the balance weight of J_FS in (3), you empirically take the best \lambda=1e-5 during experiment. To me, the scale of optimal \lambda is such small value maybe because the order of J_FS is improperly determined. My suggestion is, could you try using square root on the current J_FS, or using standard deviation of intra-segment outputs. The reasons are, first, minimizing std is a more interpretable penalty on diversion in a same segment; second, since you have used mean of outputs in J_ODM, then it is better to use a same dimension statistics, such as std of outputs in J_FS rather than sum of squared differences, when you combine J_ODM and J_FS in a uniform cost.

3.	What is the time complexity of running a comparable supervised speech recognition task with unsupervised learning method?

Minor issues:
Maybe it is a typo that the second term of Eqn (2) should be “-p_\theta(y_(t+1)=y|x_(t+1))” instead? Since the p_\theta is defined as posterior probability of the frame label given the corresponding input.

---

> ### Author Response · Authors · 2018-11-16
> **Thank you for your constructive feedback and comments**
>
> We thank the reviewer for the constructive feedback!
>
> [Top-10000 of P_LM] We found that among the 48^5 elements in the P_LM only 69553 of them are non-zero. We also found that the top 10000 elements of P_LM account for about 48.7% of the total probability in P_LM. That is, this extremely small portion of the elements in P_LM occupy almost half of the total probability.
>
> [Balance weight of J_FS] As suggested by the reviewer, we conducted experiments by taking the square root of J_FS and obtain results for different values of \lambda. We found that the best value of \lambda becomes 1e-6, which is even smaller than the original value of 1e-5. The possible reasons are explained below. First, we observe that the value of J_FS is in fact much smaller than J_ODM  (e.g., 0.15 vs 6.21). When taking the square root of J_FS, it becomes larger and we will need a smaller lambda to balance it. Second, the reason that we do not need a large lambda for J_FS is that it mainly plays the role of regularization. Ideally, if we sample all possible trajectories of \tau in (1) and match their predicted output distribution to P_LM, then the prediction within each segment would also be close to each other. However, the number of all the possible trajectories \tau in S_1 x S_2 x … S_K is exponentially large, and we cannot sample all of them in our training. Therefore, J_FS would play the role of a regularization that helps promote the consistency in the intro-segment predictions. For this reason, the regularization term J_FS does not need to be too large in practice.
>
> [Training time of unsupervised learning] In Figure 2(a), we show a learning curve of the validation error over training time when the segmentation is given by a supervised oracle. We can see that the total training could be completed in about an hour, which is similar to supervised learning. With unsupervised segmentation boundaries, it takes a longer time to converge, which is usually 4-5 hours.
>
> [Typos and minor issues] We have fixed the typos in Eq (2).

---

### Official Review · AnonReviewer2 · 2018-11-05
**Novel unsupervised cost function for phoneme recognition with promising results**

**Rating:** 7
**Confidence:** 4

**Review:**

This paper presents a method to learn an acoustic model for phoneme recognition with only the training input acoustic features and a pretrained phoneme LM. This is done by matching the output phoneme sequence distribution of the training set with the phoneme LM distribution. The cost function is proposed by extending a previously proposed unsupervised cost function (Empirical-ODM) to the segmental level, and integrating an intra-segment cost function to encourage the frame-wise output distribution to be similar to each other within a segment. The authors conducted thorough experiments on TIMIT phoneme recognition and demonstrated impressive results.

The paper is technically sound and the presentation is generally clear. The idea is interesting and novel by extending a previous unsupervised sequence modeling approach to speech recognition and exploiting the segmental structure of the problem. Unsupervised learning is an important research topic, and its application to potentially save high cost of human labeling for developing ASR systems is important to the community.

Here are a few general comments/questions:

1. It would be interesting to see whether and how much using a larger acoustic training set and a phoneme LM trained on more data can close the gap between unsupervised and supervised performance. Also it would be great to see how well the learned acoustic model performs in a full ASR system together with the lexicon and LM to predict words, which could generate more accurate unsupervised transcript than the acoustic model itself for refining the model further. These could be done in future work.

2. The current cost function is based on matching the N-gram distribution in the phoneme LM and that in the DNN acoustic model output of the training set, where N is relatively small. How could the framework be extended for the state-of-the-art LM and AM with RNNs where the history is arbitrarily long?

3. Why can this paper just use a larger mini-batch size to alleviate the effect that SGD is intrinsically biased for the Empirical-ODM functional form, while Liu et al. 2017 needed to propose the Stochastic Primal-Dual Gradient approach?

4. The paper compares the unsupervised cost function with the supervised cross-entropy function in terms of quality. How about training time? The computation looks expensive for the unsupervised case since it needs to go through all possible N-grams (which is approximated by the most frequent 10000 5-grams according to Appendix B but still a large space).

5. If the segmentation quality affects the learned acoustic model quality, why not also report the segmentation accuracy for all unsupervised systems and iterations, including the Wang et al. 2017 system?

More specific comments:

7. The outer summation in Eq(1) seems to indicate summing over all possible \tau, which is infeasible. Please clarify how it is computed.
8. Eq(5): "p(y_t | y_1 ... y_t)" should be "p(y_t | y_1 ... y_{t-1})".
9. Why are there periodic spikes in both self-validation loss and validation FER in Figure 2(a)? What training stage do they correspond to?
10. In Figure 2, "validation error" in the y-axis should probably be "validation FER". In Figure 2(b), the number ranges on the left and right of the y-axis were probably swapped.
11. Section 2.5: why is Eq(1) instead of Eq(3) used for the self-validation loss?
12. Conclusion: "the a potential" -> "the potential".

---

> ### Author Response · Authors · 2018-11-16
> **Thank you for your constructive feedback and comments**
>
> We thank the reviewer for the constructive feedback!
>
> [Future work] As suggested by the reviewer, in future work, we intend to extend our method to large-scale ASR tasks (e.g., Switchboard), and evaluate the performance in word error rate. We will also evaluate how much a larger dataset could close the performance gap between the supervised and unsupervised methods.
>
> [Extension to RNN LM & AM] We use N-gram phoneme LM because it is simple to implement and it achieves satisfactory performance. To extend to RNN-LM, we need to develop an effective way of computing the sum over z (i.e., different N-grams) in (1). For RNN-LM, the probability for each N-gram is not explicitly given. Instead, we need to score it recursively: we can treat each N-gram as a length-N sequence and use RNN to score its log-likelihood, which, after taking the exponential, becomes the probability of the N-gram. In addition, we may use beam search to pick a subset of N-grams with the corresponding probabilities scored by RNN to compute the sum in (1) approximately. For RNN-AM, we could replace the current DNN acoustic model by RNN, which will generate output distribution at each frame, then we can apply the same objective function (1).
>
> [SGD with large mini-batch] We use mini-batch SGD by dynamically increasing the batch-size. We observe that this optimization strategy empirically converges much faster than the Stochastic Primal-Dual Gradient (SPDG) and reaches a better converging point in our experiments. The reason that Liu et al. cannot reach satisfactory result by SGD may be that they did not dynamically increase the batch size during training and the difference in dataset statistics.
>
> [Training time] In Figure 2(a), we show a learning curve of the validation error over training time when the segmentation is given by a supervised oracle. We can see that the total training could be completed in about an hour, which is similar to supervised learning. With unsupervised segmentation boundaries, it takes a longer time to converge, which is usually 4-5 hours. As pointed out by the reviewer, the computation complexity for estimating a stochastic gradient is O(10000) for the sum over z in (1). However, since we use a very large mini-batch (about 20K), this computation complexity is amortized by a large mini-batch, i.e., the per sample complexity is low. In addition, this computation could also be highly parallelized in GPU.
>
> [Segmentation quality] As suggested by the reviewer, we evaluate the precision, recall, F-score and R-value of our refined boundaries and compare them to the baselines (including Wang et al. 2017). Please refer to Table 3 of our revised paper for the full results. Below, we list the results of our method and the results from Wang et al. 2017. We can see that by refining the initial boundaries provided by (Wang et al. 2017), we improve the quality of the segmentation boundaries, which leads to better PER.
>
> 		        Initial boundary (Wang et al.)		Our refined boundaries
> R-value	        82.6					                        84.8
> F-score	        80.1					                        82.6
> Recall		78.2					                        80.9
> Precision	82.3					                        84.3
>
> [sum over \tau] Since we are learning our model by SGD, we estimate our stochastic gradient by sampling this sum over \tau. Specifically, we will randomly sample one \tau at the beginning of each epoch during the training process.
>
> [Periodic spikes in Figure 2(a)] We train the model according to a fixed schedule of hyperparameters. In Figure 2(a), the mini-batch sizes gradually increase from 5000 to 20000 and the softmax temperature is decreased from 0.8 to 0.5 (i.e., gradually becomes sharper). Furthermore, in each stage, the learning rate also decays from an initial value. When the next stage begins (i.e., the position at the spikes in the figure), the learning rate will revert back to its initial value. Therefore, we believe the spikes come from the sudden increase in the learning rate.
>
> [Choice of self-validation loss] We use Eq (1) instead of Eq (3) as the self-validation loss because we will also need to tune the hyperparameter \lambda and the cost in (3) depends on \lambda.
>
> [Typos] We have fixed the typos in Eq (5), Figure 2 and Conclusion.

---

### Author Response · Authors · 2018-11-17
**General response to the reviewers**

We would like to thank all the anonymous reviewers for their helpful and constructive comments. We have provided detailed responses to each reviewer's comments and revised the paper based on their feedback. Here we summarize the changes in the newly uploaded paper.

1. We added an experiment on Unsupervised Phoneme Segmentation comparing our refined segmentation results against existing unsupervised segmentation methods.
2. We expanded our related work section greatly and discussed more related work regarding "zero-resource" speech processing.
3. We have adjusted our claim regarding the first unsupervised speech recognition.
4. We added more details on the implementation and data statistics (that are asked by the reviewers) in the supplementary material to enhance the clarity and reproducibility of our work.
5. We fixed all the typological errors mentioned by the reviewers.

---

### Meta-Review · Area_Chair1 · 2018-12-14
**Novel formulation with strong results**

**Confidence:** 5
**Recommendation:** Accept (Poster)

**Metareview:**

This paper is about unsupervised learning for ASR, by matching the acoustic distribution, learned unsupervisedly, with a prior phone-lm distribution. Overall, the results look good on TIMIT. Reviewers agree that this is a well written paper and that it has interesting results.

Strengths
- Novel formulation for unsupervised ASR, and a non-trivial extension to previously proposed unsupervised classification to segmental level.
- Well written, with strong results. Improved results and analysis based on review feedback.

Weaknesses
- Results are on TIMIT -- a small phone recognition task.
- Unclear how it extends to large vocabulary ASR tasks, and tasks that have large scale training data, and RNNs that may learn implicit LMs. The authors propose to deal with this in future work.

Overall, the reviewers agree that this is an excellent contribution with strong results. Therefore, it is recommended that the paper be accepted.